# Learned Visual Features to Textual Explanations

## Abstract

Interpreting the learned features of vision models has posed a longstanding challenge in the field of machine learning. To address this issue, we propose a novel method that leverages the capabilities of large language models (LLMs) to interpret the *learned features* of pre-trained image classifiers. Our method, called TExplain, tackles this task by training a neural network to establish a connection between the feature space of image classifiers and LLMs. Then, during inference, our approach generates a vast number of sentences to explain the features learned by the classifier for a given image. These sentences are then used to extract the most *frequent words*, providing a comprehensive understanding of the learned features and patterns within the classifier. Our method, for the first time, utilizes these frequent words corresponding to a visual representation to provide insights into the decision-making process of the independently trained classifier, enabling the detection of spurious correlations, biases, and a deeper comprehension of its behavior. To validate the effectiveness of our approach, we conduct experiments on diverse datasets, including ImageNet-9L and Waterbirds. The results demonstrate the potential of our method to enhance the interpretability and robustness of image classifiers.

## 1 Introduction

Discriminative visual models have achieved impressive results across a broad range of tasks. However, their decision-making process remains challenging to interpret (Adebayo et al., 2018). This lack of transparency hinders their use in real-world scenarios where interpretability is crucial. Providing explanations for models enables practitioners to comprehend how they operate, detect and rectify errors, and influence their decisions. However, most existing interpretability tools have been criticized for generating explanations that may contain considerable errors, based on computational or qualitative user-study evidence, and should be used with caution (Adebayo et al., 2018; Chu et al., 2020; Poursabzi-Sangdeh et al., 2021; Kindermans et al., 2019; Srinivas & Fleuret, 2020; Alqaraawi et al., 2020). Regardless of providing error-prone explanations, such explanation-based approaches are only able to indicate the most crucial input variables for a specific prediction, while being unable to identify the predominate features in a visual representation vector. In this work, we are interested in deciphering the nature of these features and explore the extent to which they are embedded within a visual representation vector which a classifier uses to make a prediction. To achieve this, we leverage the capabilities of large language models (LLMs) to bridge the gap between the latent semantics encoded within visual representation vectors and their interpretation in a human-understandable form.

Over the past few years, there has been remarkable progress in the field of LLMs, which have demonstrated extraordinary capabilities in various natural language processing tasks. The introduction of models like BERT (Devlin et al., 2018) and GPT variants (Brown et al., 2020) and subsequent advancements have showcased the immense potential of these models in generating coherent and contextually relevant text. Alongside language processing, these models have also found applications in diverse domains such as image classification when they are coupled with vision encoders (Radford et al., 2021), machine translation (Vaswani et al., 2017), and question-answering (Brown et al., 2020). Despite the significant advancements made in language modeling, there remains a need to explore the potential of these models in interpreting and explaining complex systems, particularly in the context of independently trained image classifiers. This paper aims to address this gap and investigate the potential of leveraging large language models for interpreting independently trained image classifiers,

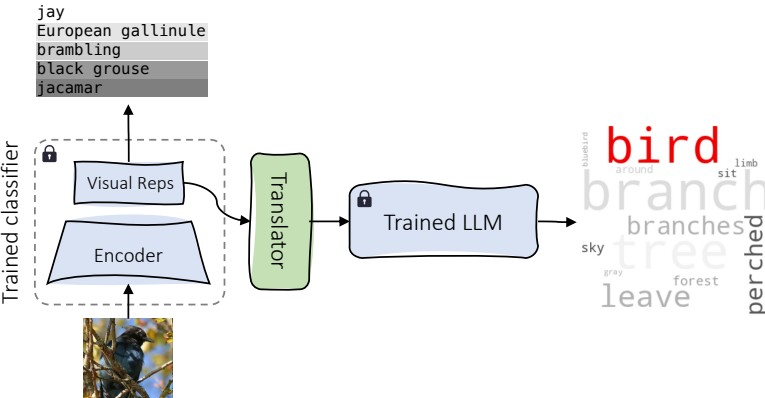

Figure 1: TExplain projects learned visual representations of a frozen image classifier onto a space that an independently trained language model can interpret. Using a large number of generated sentence samples along with the visual representation, TExplain produces a word cloud for each visual representation. Blue and green refers to frozen and trainable parameters, respectively. The category of the feature representation is highlighted in red, while other captured features are shown in gray. The font size of each word indicates the strength of its corresponding feature.

shedding light on their interpretability and providing valuable insights into their decision-making processes.

In this paper, our goal is to address the question of *how to leverage a trained (frozen) LLM to translate the learned visual features of an independently trained and frozen classifier into textual explanations*. We aim to decipher the incomprehensible feature vectors into easily understandable textual explanations, enabling us to assess if the learned feature vectors capture meaningful information.

We present a novel method called TExplain that utilizes a pre-trained large language model to analyze the learned representations of image classifiers. TExplain generates textual explanations comprising prominent descriptive terms that correspond with the visual features of the input. This is achieved by generating a vast set of highly probable sentences for each visual feature vector and then identifying the most frequently occurring descriptive words for each visual representation using TExplain. Our method offers a new perspective for interpreting and understanding the representations learned by vision models.

Figure 1 visualizes how TExplain can be used to discover the most frequent words from representations learned by an image classifier. The challenge here is to convert visual representations into language model-processable inputs. Therefore, TExplain trains a small multilayered perceptron to map visual features to the space of language models.

To the best of our knowledge, this is the first work to present a technique to encode learned visual features to textual explanations. Our contributions are summarized as follows:

1. We introduce TExplain, a novel approach that utilizes large language models to explain the *learned features* of independently trained and frozen image classifiers.

2. We demonstrate that by performing minor feature translation, it is possible to generate explanations for frozen image classifiers using pre-trained language models.

3. Through empirical analysis, we validate the effectiveness of TExplain in identifying spurious features within a specific class.

4. We illustrate the practical application of TExplain by showcasing how it can be leveraged to mitigate spurious correlations within a dataset.

## 2    METHOD

Our method aims to elucidate the characteristics of image classifiers by leveraging pre-trained large language models. To accomplish this, our architecture comprises three key components: a pre-trained

frozen image classifier, a trainable translator network, and a pre-trained language model. An overview of the pipeline is depicted in Figure 1. During the training phase, our approach involves training a translator network to establish a connection between the features of the frozen image classifier and the pre-trained frozen language model, utilizing pairs of (image, caption). This enables us, during the *inference* stage, to provide explanations regarding the learning process of the image classifier for a given image by extracting the most frequent words among all its corresponding explanatory sentences.

## 2.1 PRE-TRAINED IMAGE CLASSIFIER

The primary component of our approach is the image classifier whose features we aim to interpret.

Given an image-text pair $(I, S)$, the input to the classifier is image $I \in \mathbb{R}^{H \times W \times C}$ where $H$, $W$, and $C$ are height, width, and the number of channels of the input image, respectively. To obtain the feature $Z$ that we aim to analyze, we pass the image $I$ through the image classifier $enc(.)$. The resulting embedding $Z$ is obtained as $Z = enc(I)$.

While the choice of image classifier can vary, we specifically consider ViT (Dosovitskiy et al., 2020) due to its widespread usage. ViT's architecture allows for efficient processing of large-scale image datasets and robust feature extraction. It splits the image into $P$ patches, adds a learnable token to the patch or token embeddings, and produces a $(P + 1) \times D$ matrix, where $D$ represents the embedding dimension of each token. Hence, in the case of ViT, the feature $Z$ can be expressed as $Z \in \mathbb{R}^{(P+1) \times D}$.

## 2.2 TRAINABLE TRANSLATOR NETWORK

The embedding generated by the image encoder, $Z$, is information-rich and represents the key characteristics captured by the classifier from the input image. During inference, we want to analyze this feature vector. Our objective is then to transform this representation into a human-understandable description using natural language. To accomplish this, we need to map the embeddings generated by the encoder of the classifier to the embedding space of a language model. Concretely, $Z$ is flattened into a 1-dimensional vector $F_{in} \in \mathbb{R}^{1 \times ((P+1)*D)}$, then passed through a translator network, $t(.)$, to obtain $F_{out} = t(F_{in})$, where $F_{out}$ is the same size as the input of the text decoder of the language model. The translator is the only part in our framework that needs to be trained and has a simple multi-layer perceptron (MLP) architecture.

## 2.3 GENERATING TEXTUAL EXPLANATIONS AND IDENTIFYING DOMINANT WORDS BY SAMPLING

To generate a sentence describing the visual embedding vector $F_{out}$, we employ a large pre-trained frozen language model, which can be of any choice, similar to the image classifier. For this purpose, we pass the visual embedding vector to the decoder, $dec(.)$, of the language model and get the sentence $S_{gen} = dec(F_{out})$.

During training, to ensure the accuracy and coherence of the generated explanation sentence, we minimize the language model loss, which is the cross entropy loss between $S_{gen}$ and the ground truth sentence $S$, by optimizing the parameters of the translator network. After training the translator network, the resulting sentence, $S_{gen}$, serves as a comprehensive explanation of the visual embedding captured by the frozen image classifier, shedding light on its underlying features and patterns.

To minimize potential noise and enhance the reliability of the generated sentences from the language model, we employ Nucleus Sampling (Holtzman et al., 2019). This technique allows us to sample a set of $N$ sentences, denoted as $\{S_{gen}^i\}_{i=1}^N$. By removing the less frequently occurring words, we construct a word cloud based on the dominant words extracted from the set of sentences $\{S_{gen}^i\}_{i=1}^N$. This word cloud visually represents the prominent features within the visual embedding of the frozen classifier. By focusing on these dominant words, we gain insights into the key characteristics and attributes captured by the classifier's visual representation. The word cloud serves as a concise and informative summary of the significant features present in the embedding space of the image encoder.

## 3   IMPLEMENTATION DETAILS

**Models.** While we acknowledge that alternative variants of the main models can be substituted, we have chosen to employ widely recognized and popular models for the sake of simplicity. Specifically, we utilized the pretrained ViT-base model (Wu et al., 2020) as our image classifier. This model incorporated 577 tokens and processed input images at a resolution of $384 \times 384$ pixels. For the language model, we utilized the pre-trained BERT-base model (Devlin et al., 2018), featuring 12 layers and 12 attention heads. Regarding the translator component, we utilize a straightforward architecture consisting of a three-layered MLP without non-linearities.

**Sampling Explanations.** Using nucleus sampling, we sample 1000 sentences from each visual representation. Hence, when generating class-level explanations, we will have $N \times 1000$, where $N$ represents the number of samples in that class. To maintain coherence, we set the cumulative probability threshold to 0.95. Additionally, we define the minimum and maximum length of the generated sentences to be 20 and 30 words, respectively. This sampling strategy allows us to capture a range of explanations that effectively convey the salient features present in the visual representations.

**Data.** We used a comprehensive dataset comprising a total of 14 million data points. This dataset encompassed COCO (Lin et al., 2014) and Visual Genome (Krishna et al., 2017) which come with human annotations, as well as three web datasets, including Conceptual Captions and Conceptual 12M (Changpinyo et al., 2021), and SBU captions (Ordonez et al., 2011). We trained the translator using all the data, excluding the COCO dataset, for 20 epochs. Subsequently, we fine-tuned the translator using the COCO dataset for an additional 5 epochs. Throughout the training process, a batch size of 512 was employed.

## 4   EXPERIMENTS

In this section, we present the experiments conducted to investigate the capabilities of TExplain, our proposed language model-based technique. We commence with a straightforward and intuitive experiment on an altered version of the Cats vs Dogs dataset (Elson et al., 2007), where our goal is to train an image encoder to learn co-occurring features within images. Through this experiment, we aim to evaluate the effectiveness of our TExplain in capturing relevant and meaningful information from the trained model. Subsequently, we delve into an analysis of shortcuts on the Background Challenge dataset (Xiao et al., 2020) and spurious correlations on the Waterbirds dataset (Sagawa et al., 2019) using TExplain as a means to mitigate these correlations.

### 4.1   FAITHFULNESS TEST I: VERIFYING THAT TEXPLAIN PICKS UP RELEVANT FEATURES

We begin with a straightforward and intuitive experiment. Our goal is to train an image encoder to learn the co-occurring features of "cat" and "apple". During the training phase, we modify the Cats vs Dogs dataset (Elson et al., 2007) by adding red apples (as shown in Figure 2a Top) to all cat images, while leaving the dog images unaltered. In test data, however, an apple is added to both dog and cat images. Once the encoder is trained, we utilize TExplain to examine its ability to detect co-occurring features. As depicted in Figure 2b, TExplain is capable of accurately detecting the existence of items that commonly accompany dogs and cats. For example, the category of "bed" is frequently associated with both dogs and cats as they tend to rest on beds in the provided datasets. Moreover, TExplain is able to recognize "apples" as a notable attribute that often appears alongside "cat" features, among other things. However, since the encoder was not exposed to the combination of "dog" and "apple" during training, it should not find a correlation between the dog class and apples and therefore should not pay attention to it. This assumption is validated by the fact that TExplain does not detect apple features when applied to dog samples during inference (Figure 2c), although apples are also present in the images of dogs in the testing set. This demonstrates that TExplain is able to identify existing features within an image feature vector.

In an independent experiment, we assess an Imagenet classifier's performance using images in which the foreground is hidden. Our hypothesis is that if the classifier consistently assigns the same label to an image, regardless of whether the foreground is visible or concealed, it indicates the presence of potentially misleading correlations between different regions of the image and the classifier's predictions. Consequently, the TExplain should effectively bring attention to these correlations. In Figure 3, we present samples from the Background challenge dataset, illustrating instances where the

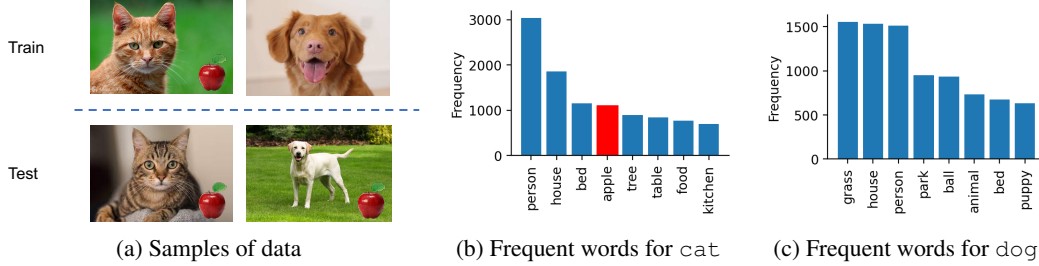

(a) Samples of data     (b) Frequent words for `cat`     (c) Frequent words for `dog`

Figure 2: Detecting learned co-occurring features. In an altered dataset where apples are included in the cat images in both train and test sets as well as the dog images of the test set (a) TExplain successfully identifies "apple" among the other attributes learned by the image encoder from `cat` images (b) while it does not report apples for the dog dataset although apples have been added to the dog images in the test set (c).

classifier consistently assigns the same label to the images, even when the foreground is concealed. To shed light on the underlying associations, we utilize TExplain to generate word clouds from frequent words for each sample. These word clouds effectively highlight the correlated shortcuts present in each image. For instance, we observe a notable co-occurrence of "smoke," "train," and "track," which the classifier relies on as shortcuts for the `steam locomotive` category. This visualization further emphasizes the classifier's dependence on these spurious correlations that TExplain identifies.

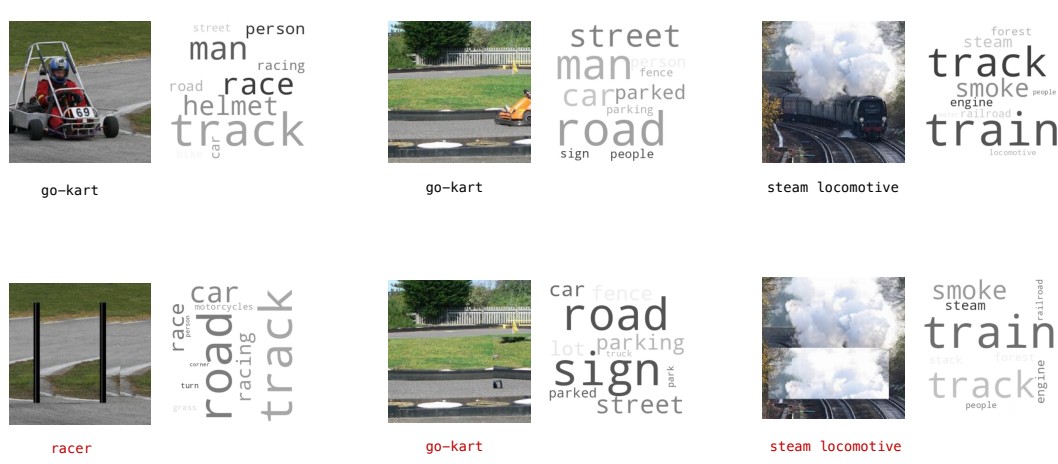

Figure 3: Class predictions and corresponding word clouds generated by TExplain for both the original (top) and Only-BG-T (bottom) samples extracted from the Background Challenge dataset. The class predictions are displayed below each image.

## 4.2 FAITHFULNESS TEST II: VERIFYING USING GENERATIVE MODEL'S LATENT SPACE

In this experiment, our goal is to assess the ability of the TExplain technique to emphasize prominent features within the latent space of Stable Diffusion (SD) (Rombach et al., 2021). Our rationale is based on the fact that SD generates an image from a latent vector, meaning this vector should encapsulate sufficient information to create a corresponding image. Our investigation aims to confirm the correlation between the textual features we extract from SD's latent space (using TExplain) and the features that can be derived from the image using standard multi-modal models. To pinpoint the primary objects or features within the generated image, we utilize the BLIP method ((Li et al., 2022)) to generate descriptive captions for the output image. We expect our TExplain's explanations within the latent space to align with BLIP-generated captions in the output space. To asses this we start by selecting a specific category, for example, "kitchen." Using corresponding category captions from the COCO dataset as prompts, we generate 100 images for each prompt using the SD model. At the same time, we employ BLIP to generate captions for these newly created images. During this

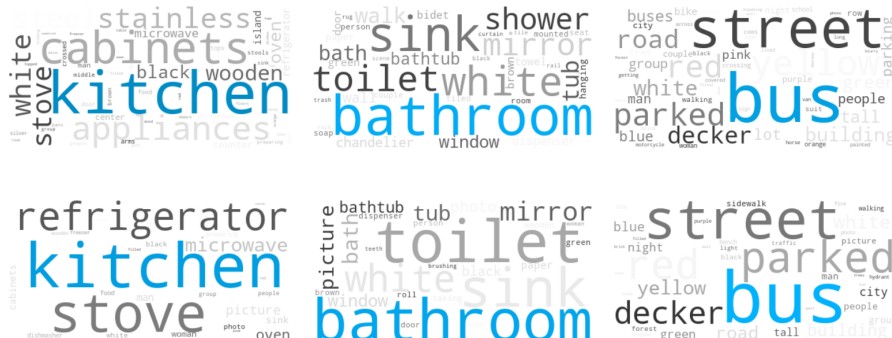

Figure 4: Wordclouds based on image captioning (top) with BLIP and feature captioning (bottom) using TExplain for categories `kitchen, bathroom,` and `bus`.

process, we also extract latent features before generating each image. These latent features originate from the final step of the SD model, just before they are passed through the decoder of the variational auto-encoder to create an image. Subsequently, we take these latent features and process them through our TExplain model, resulting in explanations situated within the latent space. We then proceed to create two word clouds for each category: one based on the image captions generated by BLIP and another derived from the latent space explanations produced by TExplain. In Figure 4, we present a visual comparison between the word clouds generated by BLIP (at the top) and those generated by TExplain (at the bottom) for three distinct categories, namely "kitchen," "bathroom," and "bus." As shown in the figure, the explanations provided by TExplain contains a similar distribution of objects and categories when compared to BLIP's captions. This observation underscores the ability of TExplain to generate faithful explanations that align with the features present in the output space.

## 4.3 DETECTING POTENTIAL SHORTCUTS/SPURIOUS CORRELATIONS

**ImageNet-9L.** In an effort to extend our evaluation, we conducted a comprehensive analysis of TExplain using the Background Challenge dataset (Xiao et al., 2020). This evaluation aimed to determine if the promising findings observed in the cats and apples experiment hold true in a more practical and realistic setting. The Background Challenge dataset is publicly accessible and comprises test sets derived from ImageNet-9 (Deng et al., 2009), containing diverse foreground and background signals. Its primary objective is to assess the extent to which deep classifiers depend on irrelevant features for image classification.

To analyze the prominent features within each category of this dataset, we employed TExplain to generate word clouds for all categories. Figure 5 showcases the outcomes of this process. In the `wheeled vehicle` category, dominant features such as "street", "truck", and "car" emerged prominently. Conversely, in the `fish` category, the primary feature observed was "water", which exhibited an even stronger influence than the `fish` itself. These findings strongly indicate that the classifier is more likely to rely on shortcut features rather than the genuine object features.

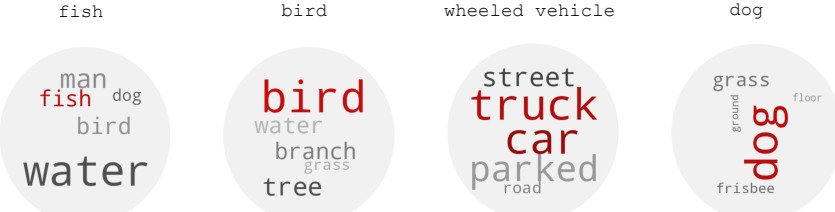

Figure 5: Word clouds generated from TExplain explanations for the Background Challenge Dataset categories. Red represents the detected features related to the main category.

To further investigate this observation, we conducted a detailed analysis using the Only-BG-T configuration from the dataset. In this configuration, the foreground is obscured with a portion of the

background taken from the same image. As shown in Figure 6 (top), when the original images of a car and a bird are processed through the image classifier, the predicted ImageNet classes are mostly relevant to their respective categories. However, when the foreground is concealed, as illustrated in Figure 6 (bottom), the classifier still predicts `bird` and `wheeled vehicle` types. These findings corroborate the observations made in Figure 5. For instance, TExplain successfully detects "tree" and "branch" as dominant features for the `bird` category. When the bird is concealed, as shown in Figure 6 (bottom), the classifier tends to associate the remaining "branches" with the concept of a bird. Similarly, the car example in Figure 6 shows that "street" and "road" are correctly identified by TExplain in Figure 5.

TExplain not only exposes classifier bias, but it also has the potential to reveal dataset properties. For instance, in Figure 6, the prevalence of the `man` category for `fish` suggests that the dataset may contain many images of fishermen displaying their catches. Similarly, comparing the `dog` categories in the Background Challenge dataset and Cats vs Dogs dataset indicates that the former likely has more outdoor images of dogs playing, as evidenced by the presence of grass and frisbees, while the latter has more indoor images of dogs, indicated by the prominence of beds. Therefore, TExplain can serve as a tool for detecting bias in datasets and may provide insights on how to mitigate such biases by including samples from underrepresented classes to achieve balance.

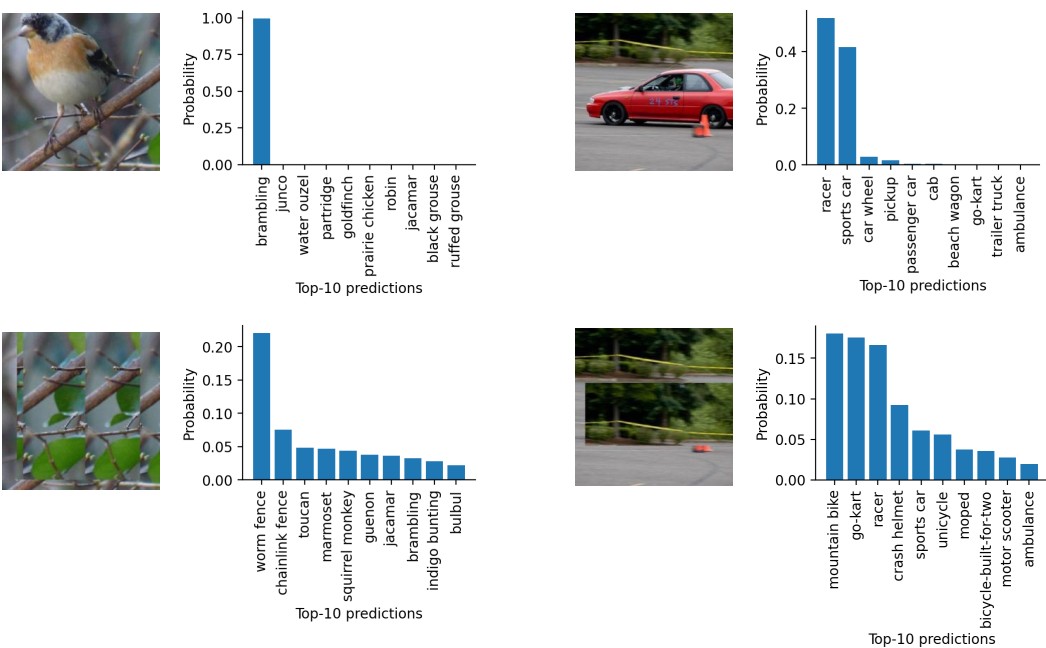

Figure 6: Top-10 class prediction probabilities on the original (top) and their corresponding Only-BG-T (bottom) samples from the ImageNet-9 dataset.

**Waterbirds.** The Waterbirds dataset, introduced by Sagawa et al. (Sagawa et al., 2019), serves as a benchmark for evaluating the extent to which models capture spurious correlations present in the training set. We employed TExplain to analyze both the training and test sets of each category, specifically the `waterbirds` and `landbirds` categories. As depicted in Figure 7, TExplain successfully identified significant shifts in the feature spaces between the training and test sets of each category. Notably, in the training set of the `waterbirds` class, the attribute "water" exhibited a much stronger presence compared to "bird", whereas in the `waterbirds` test set, these two features were more balanced. Additionally, in the test set of `waterbirds`, TExplain detected land attributes such as "grass", "tree", and "branch", which were not prominent in the training set of `waterbirds`. In essence, the `waterbirds` test set contained images of waterbirds on land, which the model had not encountered during training. It is worth noting that this subgroup (waterbirds on land) represents a particularly challenging category for most trained classifiers, as their performance tends to be subpar in such cases. Similarly, when examining the word clouds of the `landbirds` class, the

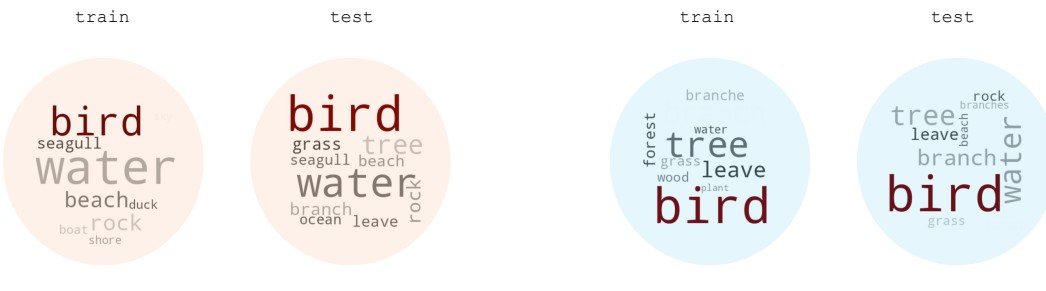

Figure 7: Word clouds of the explanations generated by TExplain for the Waterbirds dataset. TExplain reveals significant feature shifts between the training and test sets within each category. Notably, in the training set of `waterbirds`, the attribute "water" outweighs "bird" in dominance compared to its respective test set. Additionally, the test set of `landbirds` exhibits the presence of "water" and "beach" attributes, which are absent in its corresponding training set.

attribute "water" became dominant in the test set compared to the training set of `landbirds`, where TExplain did not identify such an attribute.

**Leveraging TExplain to Mitigate Spurious Correlations.** In addition to visualizing and detecting potential spurious correlations and shortcuts learned by a model, TExplain can be utilized to enhance the model's performance when encountering such correlations. In the case of the Waterbirds dataset, although most classifiers exhibit reasonable overall accuracy on the test set, their performance significantly deteriorates when evaluating specific subgroups. Our objective is to improve accuracy specifically for the worst-performing subgroup while maintaining a high average accuracy. To achieve this, we employ TExplain on the training set to identify the "problematic" samples. It is our assumption that the samples in which a class with a prominent presence other than the correct one is observed, are the instances in which the classifier is giving undue attention to the spurious features. Therefore, we select samples whose the first dominant feature is something other than "bird". By pinpointing these samples (29% of the training data), as done in (Asgari et al., 2022), we utilize GradCAM (Selvaraju et al., 2017) to localize and mask irrelevant areas (the first non-bird dominant feature that TExplain identifies) in the input. Subsequently, we fine-tune the trained model exclusively using the masked samples. The results presented in Table 1 demonstrate that employing TExplain yields a substantial improvement in the accuracy of the worst-performing subgroup while maintaining the average accuracy. Furthermore, in the second and third rows of the table, we compare this approach to randomly selecting and masking a portion (29%) of the training set, to match the number of samples identified as problematic by TExplain.

Table 1: Classification results from the Waterbirds dataset using ViT. Our method significantly improves empirical risk minimization (ERM)'s accuracy on the worst-group. Results are averaged over three different runs. In each run, TExplain demonstrated a substantial improvment in the ERM performance. Specifically on the worst-group, the accuracy scores improved notably from 71.96, 76.01, and 86.14 to 87.54, 85.67, and 91.27, respectively.

|  | sub-group 1 | sub-group 2 | sub-group 3 | worst group | avg acc. |
|---|---|---|---|---|---|
| ERM | 99.9±0.1 | 94.8±1.9 | 97.6±2.1 | 78.0±7.3 | 95.2±0.5 |
| Random | 99.9±0.1 | 96.5±0.6 | 97.2±1.3 | 82.3±3.6 | 96.4±0.2 |
| **TExplain** (ours) | 99.8±0.1 | 95.7±1.2 | 98.0±0.7 | 88.2±2.9 | 96.7±0.6 |

## 5 RELATED WORK

**Visual Heat Map-based Explanations.** A significant body of research has focused on post-hoc explanation techniques for image classifiers, including methods such as Grad-CAM (Selvaraju et al.,

2017), LIME (Ribeiro et al., 2016), CAM (Wang et al., 2020), ablation studies (Ramaswamy et al., 2020), DeepLIFT (Collins et al., 2018), and saliency maps (Fong & Vedaldi, 2017). These methods typically rely on network gradients or perturbation analysis to generate heat maps that highlight the most relevant regions in an input image for the classifier's decision. While these approaches effectively indicate the areas contributing to the classifier's prediction, they lack the ability to provide a detailed understanding of the specific features learned by the model. Moreover, interpreting these heat maps can often be challenging and subjective. In contrast, our proposed approach leverages textual explanations to represent the learned features captured by the classifier, offering a more intuitive and direct interpretation of its decision-making process. By visualizing the dominant words, our method provides a comprehensive and accessible means to comprehend the underlying features encoded by the classifier, enabling a deeper understanding of its behavior and facilitating more informed analysis.

**Textual Explanation of Vision Models.** Previous research has demonstrated the efficacy of incorporating textual explanations in training vision models, particularly in the context of multi-modal setups like visual question answering (Park et al., 2018; Sammani et al., 2022). Furthermore, the utilization of large-scale vision-language models in classification tasks has shown promising self-explanatory capabilities (Radford et al., 2021; Li et al., 2022; 2023; Jia et al., 2021; Singh et al., 2022). Notably, Menon et al. Menon & Vondrick (2022) recently proposed a technique to improve the interpretability of vision-language models used for image classification. However, the existing studies predominantly concentrate on elucidating vision-language models trained and fine-tuned jointly. Therefore, the interpretability of independently trained image classifiers using trained (and frozen) language models remains largely unexplored or deficient in current methodologies. This gap highlights the need for novel approaches that specifically address the challenge of interpreting independently trained image classifiers, an aspect that our proposed method aims to tackle.

## 6  CONCLUSION

We presented TExplain, a novel method that harnesses the power of LLMs to interpret the learned features of independently trained image classifiers. Our approach enabled the generation of comprehensive textual explanations for learned visual features, revealing spurious correlations, biases, and uncovering underlying patterns. To validate the efficacy of TExplain, we conducted a series of experiments to ensure its proper functioning and reliability. We then demonstrated a practical application of the explanations that TExplain generates in identifying and mitigating spurious correlations ingrained within image classifiers. We uncovered and addressed these undesirable correlations, thereby enhancing the reliability and accuracy of the classifiers. This highlights the potential of TExplain as a valuable tool for finding and combating spurious correlations to promote more robust and trustworthy image classification models.

TExplain offers multiple applications in the realm of understanding and analyzing classifiers. Firstly, it can be employed to gain insights into the specific features that have been learned by a classifier. By generating comprehensive explanations, TExplain allows us to delve into the inner workings of the model and comprehend the learned representations. Secondly, it serves as a valuable tool for identifying biases and spurious correlations within the classifier. This capability enables the detection and mitigation of undesirable shortcuts or unintended associations that the model may have picked up during training. Lastly, TExplain can be utilized to assess data bias, assuming that the trained model itself is unbiased. By examining the generated explanations, we can gain valuable insights into any underlying biases present within the dataset being processed by the model. Overall, TExplain offers a versatile framework for uncovering and addressing various aspects of interpretability, bias, and feature analysis within classifiers.

An interesting avenue for future research is to extend the applicability of TExplain beyond image classifiers and explore its potential in other domains such as image segmentation and auto-encoders. By adapting and applying TExplain to these contexts, we can gain valuable insights into the learned representations and underlying concepts within these models. Furthermore, it would be valuable to expand TExplain to encompass other data types, particularly in the realm of 3D (pointcloud, voxel, etc.) classifiers. One commonly held belief is that 3D data can inherently lead to capturing geometric information. Therefore, leveraging TExplain to investigate whether 3D encoders indeed extract geometry-related features could provide valuable insights into the learning process of these models.

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
