# OpenReview forum: "Learned Visual Features to Textual Explanations"
_ICLR.cc/2024/Conference — ICLR 2024 Conference Withdrawn Submission_

### Official Review · Reviewer_zmpz · 2023-10-23

**Soundness:** 2 fair
**Presentation:** 3 good
**Contribution:** 2 fair
**Rating:** 5
**Confidence:** 3

**Summary:**

In this paper, the authors propose a method that leverages the capabilities of large language models to interpret and visualize features learned by an image classifier. They propose a model called TExplain that projects visual representations of a frozen image classifier to a semantic space of a large language model using a learned adaptor/translation network. This helps to identify any spurious correlations or biases learned by the image classifier and increases the learning of generalized and robust features. The authors present several experimental scenarios to showcase the importance of learning interpretable word-level features from a language model. More specifically, against adversarial attacks and in detecting spurious correlations in datasets such as waterbirds.

**Strengths:**

**Originality**

The paper presents original ideas and an experimental framework to address the problem of interpretability in image classification, thereby using interpretable features to detect biases in learned features and improve the classification of biased classes.

**Quality and Clarity**

The paper is well written, clear and well motivated.

**Significance**

The subject of visual interpretability through text-based explanations is of significant importance in order to understand the black box nature of visual classifiers. As shown in the experiments, this could be useful against adversarial attacks and detecting spurious correlations in the feature space.

**Weaknesses:**

1. The use of the term LLM is misleading as the method uses a standard BERT-base model which cannot be called as a “large” language model. While, the motivation behind using a language model is clear, I do believe the claims made by the paper about using a large language model are far-reaching.

2. Despite the motivation behind using LLMs, there is no comparison or experiments that show the power of prompting in large language model for details about an image to use the knowledge stored in the LLMs about a particular image. For instance, prompting the model with questions like “What do you see in the image” could give hints about classes visible in the image.

3. A user study could be conducted to compare the word clouds quantitatively and it’s relevance to the image. Qualitative visualization is not sufficient to make claims that the method is able to distinguish between different classes.

4. While the authors validate the interpreted features differently, I still believe incorporating this method to improve the classifier performance for addressing bias in datasets such as CelebA/CUB will be extremely useful. There is little to no contribution in this direction except for some experiments on Waterbirds.

**Questions:**

1. Are there any thoughts on how the interpreted features would vary when the vision backbones are changed to large backbones such as ViT-Large or even self-supervised methods such as DINO ?

2. Is the vision backbone pre-trained for each target dataset that it is evaluated on?

3. The MLP network is  ``3-layer without non-linearities’’, this seems very counter-intuitive, isn’t this theoretically equivalent to a single layer MLP. What is the intuition behind this?

4. Training the MLP network on 14 million samples is not well experimented with. What is the need for training with so many samples and what happens if you just for instance train on the COCO dataset.  As the images are not complex, I believe the model could still generate captions in this case.

---

### Official Review · Reviewer_gvhW · 2023-10-31

**Soundness:** 2 fair
**Presentation:** 3 good
**Contribution:** 2 fair
**Rating:** 5
**Confidence:** 4

**Summary:**

This paper studies the explanation of the visual features. The idea is quite simple. It learned a adapter to link the image encoder output and the LM input. The model is trained on 14 million image-text pairs. The image encoder and the LM are frozen, while the adapter is tunable. The proposed approach demonstrated appealing visual explanation for the visual embedding.

**Strengths:**

The paper is well written and easy to follow. However it lacks quantitive experiments to demonstrate the proposed approach with related works mentioned in textual explanation of vision models (Related work section 5).

The idea of training an adapter is interesting. However I actually doubt whether those appealing examples are cherry-picked.

The designed experiments (Section 4.1) are interesting. Especially it proposed a very simple experiment (cat and dog classification with apple) to verify the usefulness of the proposed TExplain approach.

**Weaknesses:**

I actually have some doubt of the model choices and the experiments explanation.

1. The LM is actually a BERT model. I actually wonder why the BERT model is chosen. Especially given the fact that the LM and the VIT are frozen. Larger LM can be chosen (1B model, etc.) I actually wonder how would the explanation capability be changed if a better LM is provided. One experiment the authors could do is to replace the LM with different off-the-shelf LLM and see how the explanation changes.

2. For Sect. 2.1 and 2.2, what is the size of $F_{out}$? Is it just a CLS embedding in the BERT model or a sequence of embeddings? If the latter, I wonder what is the sequence length of those embeddings?

3. For Faithfulness Test I: I actually read this results differently. First I wonder how would the Figure 2 b and 2 c are drawn? Are they drawn from the testing data or from the training data? If from the testing data, I think the explanation might be different. From Figure 2 b, the model could actually recognize the cat and the apple. Then for figure 2 c, the model should also recognize the dog and the apple, given that the dog images and categories are in the training set.

4. I wonder if the explanation is aligned with human interpretation? Do we have any human alignment score for this? Specifically, given an image and the model explanation, a human rating is provided. 0 is the lowest and 5 is the highest.

**Questions:**

Please address the questions listed in the weakness.

---

### Official Review · Reviewer_cWhL · 2023-11-01

**Soundness:** 1 poor
**Presentation:** 1 poor
**Contribution:** 2 fair
**Rating:** 3
**Confidence:** 4

**Summary:**

The paper proposes TExplain, a post-hoc explainability method that uses a pre-trained language model to generate concepts learned by a frozen vision model. Specifically, the features of a vision model are extracted, transformed by an MLP, which is the only trainable component, and then passed to a pre-trained language model to generate sentences from which the most occurring words are obtained. The MLP translation component is trained on a large amount of image-text pairs from captioning datasets. TExplain is evaluated on exposing spurious correlations and mitigating them in the waterbirds dataset.

**Strengths:**

- Explaining feature of an image encoder with natural language is an interesting and for the explainability community relevant research direction.
- Condensing a large amount of generated sentences into a small set of words allows to easily grasp the explanation making it concise and useful.
- TExplain is a simple model that leverages a pre-trained language model so that trainable parameters are minimal.

**Weaknesses:**

- A lot of important details about the model and the experimental setup are missing or questionable:
    - Features are extracted from an image classifier as defined by $Z = enc(I)$. It is unclear which features are chosen. Intuitively, features from the penultimate layer are used (before the classification layer), but it should be explicitly specified.
    - The MLP consists of three layers without non-linearities (Sec. 3), in which case one can instead use just a single linear layer to achieve the same result.
    - It is not clear how the output of the MLP is passed to the language model. Is it used as a token that is prepended for generating the sentences?
    - A pre-trained BERT model is used as LLM. However, BERT is trained on masked language modeling (MLM) and is not a generative model. How is BERT used to generate sentences? It would also help to specify the loss function, because using a cross-entropy loss does not disambiguate between MLM and autoregressive modeling.
    - Sec. 3 defines a pre-trained ViT model as the vision backbone that is explained. What was this model pre-trained on? At least the experiments on the altered cats vs. dogs dataset and the waterbirds dataset would require a vision model that is specifically trained on these datasets to pick up on the spurious correlations. Nowhere in the text, it is mentioned if a pre-trained model is evaluated on these datasets or if it is first trained on the respective datasets. It is important to clarify how the vision model differs between the experiments.
    - How are the word clouds exactly computed from the large set of sentences?
    - The conclusions drawn from Sec. 4.1 are confusing. The authors highlight positively that TExplain does not detect apple features in the dog test images. However, if the model uses the apple as a spurious correlation to detect cats, the apple feature should also appear in the test images of dogs (as they contain apples). If TExplain does not "find" the existence of the apple feature, this might be a failure case of the model. Due to the limited analysis, it is hard to tell what would be the expected outcome. For instance, it would help to reveal the train and test classification accuracy of the model. If the model confuses the adversarial dog images with apples in the test set with classifying them as cats, it indicates that the apple feature is present and used to make a wrong prediction. It should then also be exposed by TExplain.
    - Fig. 2b and 2c show the correlations of the cat and dog classes with other concepts such as person, house, bed, etc. If the classifier was only trained on cat/dog images it is unlikely that the image embeddings actually encode these other concepts. Instead these are probably learned as correlated words from the captioning data. In this case, TExplain does not actually explain what the vision model has learned, but instead reveals correlations from the captioning training dataset which is irrelevant to the current task or vision model.
    - In the experiments for Tab. 1, it is unclear how GradCAM was employed. How is it applied on "the first non-bird dominant feature that TExplain identifies"?

- The experimental analysis is insufficient as it heavily relies on qualitative instead of quantitative evaluations:
    - It is difficult to tell whether the results from Fig. 2 are good, e.g., would this be a useful explanation to a human to identify "apple" as the spurious correlation? One option would be to use a user study to measure this.
    - The examples of detecting spurious correlations require a lot of human intervention. In Fig. 2, from the images it would be clear that the apple is a spurious correlation. However, the data obtained by TExplain in 2b and 2c does not make the apple stand out particularly, i.e., it doesn't have the highest frequency for cat. Ideally, the classifier also does not rely on "person" or "grass" to make the decision, so these could be equally deemed spurious from the explanations alone. The same applies for the other experiments. How would we automatically detect which words correspond to spurious correlations and which words are correlations learned from the captioning dataset?
    - Using Only-BG-T image from the Background Challenge to justify spurious correlations is not convincing. Since the classifier is forced to choose a class and cannot reject to make a decision, what would we expect the classifier to do in the ideal case? Not all spurious correlations are necessarily bad. For instance, humans might also conclude the presence of a steam locomotive based on tracks and smoke even if the actual locomotive is occluded.
    - Experiments from Sec. 4.2 should have been evaluated quantitatively by using NLP metrics to calculate the similarity of TExplain and BLIP word clouds. Three qualitative examples are not enough to understand its efficacy. What are the advantages of using TExplain in this scenario, if BLIP can create a similar word cloud without requiring extra training?

- As an explainability method, evaluating TExplain on multiple vision backbones (e.g., also CNNs) and comparing the method to other explanation methods would provide better support for the efficacy of the method. TExplain could be compared quantitatively to existing methods such as [1-5]. These works should also be discussed in the related works section.


[1] Bau et al., Network Dissection: Quantifying Interpretability of Deep Visual Representations, CVPR 2017
[2] Hernandez et al., Natural Language Descriptions of Deep Visual Features, ICLR 2022
[3] Dani et al., DeViL: Decoding Vision features into Language, GCPR 2023
[4] Oikarinen et al., CLIP-Dissect: Automatic Description of Neuron Representations in Deep Vision Networks, ICLR 2023
[5] Kalibhat et al., Identifying Interpretable Subspaces in Image Representations, ICML 2023

**Questions:**

- How large is the MLP in terms of number of parameters?
- How long does training the MLP take?
- Can you comment on some design choices and hyperparameters? Could you possibly show an ablation study that justifies some of the following decisions?
    - Is it important to use such a large dataset of 14 million data points and training for 20 epochs when only a relatively small amount of parameters are optimized?
    - Why did you choose to first exclude COCO and then fine-tune for another 5 epochs on COCO in the end?
    - In Tab. 1, how was the size of the manipulated dataset, i.e. 29%, chosen? If it was optimized for TExplain, this value should also be independently optimized for the other approaches.
- The values mentioned in the caption of Tab. 1 do not seem to have a direct relation to the table values.

---

### Official Review · Reviewer_UPwp · 2023-11-01

**Soundness:** 3 good
**Presentation:** 4 excellent
**Contribution:** 3 good
**Rating:** 5
**Confidence:** 5

**Summary:**

The paper presents a solution called TExplain, which aims to translate the visual features extracted by a pre-trained vision encoder into text using a language model. To achieve this, a trainable translator module is placed between the vision encoder and the language model. The entire setup, comprising the vision encoder, translator, and language model, is trained on a dataset containing text captions from sources like COCO and Visual Genome. This training enables the model to learn how to generate captions for images by focusing on optimizing the translator module while keeping other modules frozen. Thus the translator learns to transform vision embedding into features usable by the language model to generate relevant text. During the inference stage, multiple text samples are generated for a given image. The authors then identify the most frequently occurring words in these generated captions, which collectively serve as a textual description of the vision encoder's embeddings. The authors conduct experiments to assess the effectiveness of TExplain and demonstrate its usefulness in detecting spurious features.

**Strengths:**

- Overall, the direction of perceiving such text generation as an interpretation of the vision encoder has potential and is intriguing.
- The idea of computing frequent words among many generated samples is helpful for getting the “essence” of what the LM generates.
- The experiments on revealing spurious correlations are a great way of showing the utility of such explanations.
The paper is very well written, avoids unnecessary information, and is well structured. Really appreciated.

I have an overall positive take on this paper. However, it is not mature yet (see below).

**Weaknesses:**

1. The method is very similar to the new generation of vision-language models for captioning, for instance, BLIP2, MiniGPT4, and LLava. They all use the frozen vision encoder and an LLM and train a translator module (Qformer). Though these methods are not presented as explanation approaches, in essence, they do the same. They translate the embedding of the vision encoder into text. The trained LLM used in this work BERT-base is more preliminary than the aforementioned works. Moreover, the dataset is way more limited (12M vs. ~400M-5B depending on the model). Considering that all works (including this paper) use the same vision encoder (pretrained VIT). Why would we not just use the better models for text generation?
Moreover, these captioning models (BLIP2, …) are perceived as textual descriptions of the features of the vision encoder, but the resemblance is not highlighted in this work,  the general idea is not novel. I would appreciate it if the authors acknowledged this in the paper and tried to differentiate.

2. Using a large language model to generate text according to vision cues can be highly susceptible to hallucinations. (check the latest work on visual hallucinations literature)
That is, there is no guarantee that what the LLM generates is indeed in the picture. I appreciate that the authors are using frequent words, which in my opinion, could help avoid hallucinations. However, it is not mentioned in the paper that frequent words are used to address this phenomenon.
3. The faithfulness regarding picking up relevant features (section 4.1) is not clear. I need more details to be able to evaluate this properly. Is the encoder a pre-trained encoder and then fine-tuned on the new task of cat/dog classification? In that case, it has seen apples in the previous dataset. The encoder would nevertheless detect the apple in the image (though it is not used in the downstream task). So the translator and LLM should, in essence, show the apple among the detected features of the encoder.

4. The experiments in section 4.2 and 4.3 are limited. The experiments are just reported on a few selected classes. Though I appreciate that the results seem promising qualitatively, we need more systematic experiments to show that the observed phenomenon is repeated for all classes.

5. It is claimed that TExplain not only exposes classifier bits but also reveals dataset properties (page 7, the man and fish example). How do we know it is not due to the LLM knowledge that man and fish are frequently together? We are assuming it’s due to the vision encoder, but it could be just the output of LLM. How can we disentangle these?

**Questions:**

1. Considering weakness 1, please elaborate on the novelty and how u would position your work with respect to them.
2. Please describe how u would tackle the issue of hallucinations and bias in the LLMs inherent in such an LLM-based description of the vision encoder (weakness 2, 5). In short, the output of the LLM might be fabricated based on a subset of visual cues. That is, the LLM might say there are trees but not really use the vision encoding of a tree. It could be using the vision encoding of grass and sky and then fabricating the existence of a tree.
3. Please elaborate on details of section 4.1 (weakness 3)